# Preparation of Functionalized Magnetic Polystyrene Microspheres and Their Application in Food Safety Detection

**DOI:** 10.3390/polym15010077

**Published:** 2022-12-25

**Authors:** Xin Teng, Xingyu Ding, Zhuxin She, Yi Li, Xiaohui Xiong

**Affiliations:** College of Food Science and Light Industry, Nanjing Tech University, Nanjing 211816, China

**Keywords:** micro-suspension polymerization, iron tetraoxide, polystyrene, high performance liquid chromatography tandem mass spectrometry, rapid detection

## Abstract

Based on the specific binding of sulfonic acid groups to melamine, β-agonists and other compounds, Fe_3_O_4_ nano-magnetic beads were coated with polystyrene using an improved micro-suspension emulsion polymerization method, thus forming core–shell magnetic polystyrene microspheres (Fe_3_O_4_@PS) with Fe_3_O_4_ as the core and polystyrene as the shell. These functionalized microspheres, which can be used as magnetic solid-phase extraction (MSPE) adsorbent, were prepared after further sulfonation. These microspheres were characterized by Fourier transform infrared spectroscopy (FTIR), scanning electron microscopy (SEM), transmission electron microscopy (TEM), particle size analysis and saturation magnetization measurement. The results showed that these sulfonated magnetic polystyrene microspheres had favorable sphericity. The particle size of these microspheres ranged from 1 μm to 10 μm. Additionally, these microspheres had good dispersion and magnetic responses in both inorganic and organic solvents. Moreover, these functionalized magnetic polystyrene microspheres were tested and evaluated by high performance liquid chromatography tandem mass spectrometry (HPLC-MS/MS). The results indicated that these sulfonated magnetic polystyrene microspheres (Fe_3_O_4_@SPS) could effectively adsorb such illegal additives as β-agonists and melamine in the food matrix.

## 1. Introduction

As an important method for the prevention and control of animal diseases, veterinary drugs play a crucial role in improving animal production efficiency and livestock product quality, and maintaining ecological balance. Animal-derived foods, such as meat, eggs, aquatic products and dairy products constitute a large category of human foods that can provide vitamins, minerals, fats, and proteins for the human body [1]. Veterinary drug residues in animal-derived foods refer to the accumulation and residue of original drugs, their metabolites and related impurities in animal tissues, milk or eggs of mammals or the ecological environment after the use of veterinary drugs (including drug additives) [2]. Veterinary drug residues not only directly endanger the human body and increase drug-resistant strains, but also pose a potential threat to the environment and human health through multiple food chains [3,4]. China has a higher level of production, processing and consumption of livestock and poultry products. Ensuring the quality and safety of animal products contributes to the health of the people, the development of production and trade, and the sustained and healthy development of the economy in China. Therefore, it is important to take effective measures to control and reduce the occurrence of veterinary drug residues.

At present, veterinary drug residues and illegal additives in food are commonly detected by the immune analysis method [5], biosensor method [6], instrument analysis method [7] and other methods. Among them, instrumental analysis is mainly performed based on large, expensive and sophisticated instruments, including liquid chromatography, gas chromatography and its tandem mass spectrometry [8,9,10,11]; it is the mainstream detection method. As with the detection techniques specified in the national standard, the above methods can be adopted to obtain reliable results and can also be used to analyze the specificity of the test object under the interference of multiple substances. Owing to its high sensitivity, small sample size and small matrix interference, UPLC-MS/MS has become an advanced method for detecting veterinary drug residues [12,13].

However, veterinary drug residues are characterized by low residual levels in samples and complex sample matrices, which makes it difficult to directly determine trace and trace residues in samples. In addition, it is also difficult to obtain the required precision for veterinary drug residue detection. Furthermore, detection instruments are prone to damage [14]. Therefore, sample pretreatment technology plays a crucial role in veterinary drug residue detection. Sample pretreatment is a complex process which is mainly composed of extraction, purification and concentration [15]. In terms of the analytical methods with low detection limits, such as liquid chromatography-mass spectrometry, sample preparation is more important than the detection process, and complex pretreatment procedures are needed. Sample pretreatment technology is the key link during veterinary drug residue detection, and it directly affects analytical efficiency and accuracy. Due to their cumbersome operation, poor stability and low specificity, traditional sample preparation technologies have been replaced by some new separation and purification methods, such as solid phase extraction [16], solid phase microextraction [17], matrix dispersive solid phase extraction [18], immunoaffinity chromatography, molecular imprinting technology [19,20], supercritical fluid extraction and accelerated solvent extraction [21]. These new technologies feature a small sample size, favorable specificity and selectivity of method, high automation, simple manual operation, environmental friendliness and small dosages of organic reagents. Among them, magnetic solid phase extraction (MSPE) can be employed to purify complex food matrices, reduce the influence of matrix effect in mass spectrometry analysis, decrease the trace concentration of extremely low concentration components and convert solvents that can also be achieved by traditional solid phase extraction technology. Additionally, MSPE also outperforms traditional sample pretreatment technologies in separation and purification, and has been considered an important component in sample pretreatment methods [22,23]. In recent years, new efficient, green, and environmentally friendly magnetic adsorption materials have become a research hotspot, and these materials are also the key to the detection of veterinary drug residues.

The magnetic polymer composite microsphere is a new functional material compounded from inorganic magnetic nanoparticles and organic polymer materials. These microspheres have both the functional properties of polymer materials and the magnetic response of inorganic nanoparticles. They can attract functional groups (such as amino, carboxyl, and aldehyde) through modifying polymer materials, so as to graft enzymes, catalysts and other guest molecules [24]. At the same time, magnetic polymer composite microspheres can also realize magnetic separation and directional movement under the action of external magnetic fields. As a new functional material, magnetic polymer composite microspheres have broad application prospects in many fields, such as cell separation, targeted drug delivery, biological separation and immunoassay [25]. At present, there are many methods for the synthesis of magnetic polystyrene microspheres, such as monomer polymerization, the blending embedding method and the activation swelling method. The sample pretreatment process is shown in Figure 1. In this paper, magnetic polystyrene microspheres with favorable performance were prepared by the optimized process based on the suspension emulsion polymerization method. In addition, the characterization, functionalization and practical sample adsorption application were carried out on these magnetic polystyrene microspheres.

## 2. Materials and Methods

### 2.1. Materials

Fe_3_O_4_ was home made in the laboratory with a particle size of about 200 nm, and sulfuric acid (98%), hydrochloric acid (37%), oleic acid (99%) and chlorosulfonic acid (98%) were purchased from Shanghai Zhangyun Chemical Co.; styrene (St, 99%), divinylbenzene (DVB, 99%), benzoyl peroxide (98%), trienyl isocyanate (TAIC, 99%) were purchased from Shanghai Lingfeng Chemical Reagent Co., Ltd. (Shanghai, China); cetyl alcohol (98%) and polyvinyl alcohol 1788 (alcohol solubility: 87–89%) were purchased from Shanghai Aladdin Reagent Co.; clenbuterol hydrochloride (1000 μg/mL), ractopamine (1000 μg/mL), salbutamol (1000 μg/mL), and melamine (1000 μg/mL) were purchased from Tanmo QA National Center for Standard Materials (Beijing, China); sodium acetate (99%), sodium dodecyl sulfate (99%), polyvinyl alcohol 1788 (alcoholic solubility: 87–89%) and a neutral alumina column were purchased from Taizhou Kangzida Experimental Equipment Co., Ltd. (Xi’an, Shanxi); The styrene, divinylbenzene and trienyl isocyanate were stored in a refrigerator at 4 °C before the experiment started. Sodium dodecyl sulfate and polyethylene glycol 4000 were mixed in a certain ratio, and styrene was passed through neutral alumina to remove the polymerization inhibitor.

### 2.2. Overview of Sulfonated Magnetic Polystyrene Microspheres

The magnetic polymer microspheres can be prepared by suspension polymerization based on the following principles. In the presence of magnetic powder, suspension stabilizer and surfactant, one or more monomers are homopolymerized or copolymerized on the surface of magnetic particles under the action of initiators, and the magnetic particles are wrapped in the polymerization products. Suspension polymerization is restricted by the wide distribution of particle sizes. Micro-suspension polymerization is a reduced suspension polymerization. In this reaction, the oil phase is dispersed into small droplets and polymerized at a certain temperature by a high-speed shear device in a system composed of magnetic powder, water, monomer, oil-soluble initiator, dispersant and surfactant. The particle size of suspension polymerization products is 0.01–5 mm, with the most ranging from 0.05 mm to 2 mm. The particle size of emulsion polymerization products is about 0.05–1 μm, which is much smaller than that of suspension polymerization products. In this paper, micro-suspension polymerization was used to prepare magnetic polymer microspheres with a particle size between both. Xu [26] prepared some magnetic polystyrene microspheres with a particle size ranging from 0.5 μm to 2 μm using micro-suspension polymerization. Wang [27] synthesized magnetic polystyrene microspheres with a particle size distribution of about 20 μm using micro-suspension polymerization. Based on previous efforts, magnetic polystyrene microspheres with monodisperse, uniform particle size distribution and good magnetic response were prepared using an improved micro-suspension method. Based on the work of Omi [28] et al. we obtained satisfactory magnetic composite microspheres by optimizing the ratio of initiator and magnetic microspheres, the reaction time and the pH of the reaction system as a prerequisite for the experiments conducted in this paper.

The preparation technology of magnetic polystyrene microspheres (sulfonated magnetic polystyrene microspheres, Fe_3_O_4_@SPS) functionalized by strong acids has been improved specifically by adding concentrated sulfuric acid or other sulfates to modify polystyrene (PS) coated on the surface of these magnetic microspheres, mainly using the sulfuric acid method [29], sulfur trioxide method [30] and chlorosulfonic acid method [31].

The preparation of Fe_3_O_4_@SPS with the chlorosulfonic acid as a sulfonating agent has some advantages, including strong reaction ability, low temperature, short time, small amount of feed, high sulfonation degree and easy discharge of generated HCl [32]. Hence, it is an ideal sulfonation method. Idibie [33] et al. explored the sulfonation kinetics of chlorosulfonic acid polystyrene-butadiene rubber (PSBR) with different initial concentrations. A prediction model was established to predict the sulfonation degree at different initial concentrations of acid. Zhou [34] et al. investigated the effects of reaction temperature, time, dosage of sulfonating agents, solvent system and solvent dosage on the preparation of Fe_3_O_4_@SPS with the chlorosulfonic acid as a sulfonating agent. Luo [31] et al. prepared sulfonated porous polystyrene microspheres as heterogeneous catalysts for biodiesel using a simple two-step synthesis method. These sulfonated magnetic polystyrene microspheres had high acid density (2.611 mmol/g), good thermal stability (up to about 200 °C) and a suitable specific surface area. In this paper, the preparation of Fe_3_O_4_@SPS with the chlorosulfonic acid as a sulfonating agent was modified and optimized. By comparing the experiments, the selection of sulfonation reagent, the control of sulfonation time and the adjustment of the ratio, we finally decided to choose chlorosulfonic acid as the sulfonation agent; the molar ratio of chlorosulfonic and magnetic composite microspheres was 1:1 and the optimal sulfonation time was 5 min.

### 2.3. Preparation of Magnetic Polystyrene Microspheres

Firstly, the Fe_3_O_4_ powder was prepared by the solvothermal method [35], the oleic acid was dissolved in anhydrous ethanol, and the oleic acid–ethanol solution with a concentration of 3% was prepared. Then, 0.5 g of Fe_3_O_4_ powder was mixed with a certain concentration of the oleic acid–ethanol solution and evenly placed in a three-neck flask. Subsequently, the mixture was placed in a water bath at 55 °C for 4 h, washed and filtered alternately with anhydrous ethanol and ultrapure water, and dried in a vacuum.

The magnetic powder OA-Fe_3_O_4_ (0.5 g) modified by oleic acid coating was dispersed in 5.0 mL of styrene and 1.0 mL of divinylbenzene, and 0.5 g of benzoyl peroxide was added for dissolution for 12 h. Then, 0.5 g of triallyl isocyanate, 0.2 g of hexadecanol, 100 mL of ultrapure water, 10 mL of 5% PVA solution, and 2 mL of 5% SDS-PEG solution (the ratio of SDS to PEG4000 is 3:2) were mixed evenly in a 250 mL three-neck flask. Subsequently, the Fe_3_O_4_-St solution that had been ultrasonically dispersed for 40 min was added and stirred at a medium speed. Next, the mixture was condensed and refluxed in a water bath thermostat, followed by treatment at 70 °C for 4 h. After that, the mixture was treated at 80 °C for 3–4 h. The mixture was washed and filtered alternately with anhydrous ethanol and ultrapure water and dried in vacuum to obtain brown powder products. After being screened with a 60-mesh sieve, the mixture was soaked in 1 mol/L hydrochloric acid for 24 h, washed to a neutral state, and dried. Finally, the magnetic substances were separated by magnets.

### 2.4. Functionalization of Magnetic Polystyrene Microspheres

The dried polystyrene microspheres (1 g) were added to a three-neck flask. After these polystyrene microspheres were swollen with carbon tetrachloride (20 mL) for 3 h, chlorosulfonic acid (0.4 mL) was added under mechanical stirring. After the reaction was completed at 25 °C, the material was washed and filtered with ethanol and ultrapure water to a neutral state and then soaked overnight in water. The next day, the material was washed with ultrapure water to a neutral state, washed with ethanol many times and dried in vacuum for 24 h.

### 2.5. Determination of Adsorption Properties of Functionalized Magnetic Polystyrene Microspheres

Sample pretreatment of melamine: Samples (2 g, accurate to 0.01 g) were weighed in 15 mL scale test tubes. After 7 mL of anhydrous acetonitrile was added, the oscillation and vortex were performed for 2 min. Then, 1 mL of ethylenediamine was added to perform volume metering with ultrapure water to 10 mL, followed by oscillation and vortex for 2 min. Subsequently, ultrasonic extraction was performed for 15 min at 8000 r/min and 4 °C under freezing conditions for 8 min. The mixture was filtered for later use. Next, 2 mL of the mixture was taken under nitrogen drying at 80 °C to be re-dissolved with 1 mL of 20% methanol solution for testing on the machine.

Chromatographic conditions: The chromatographic column was the TM HILIC column (2.6 μm, 100 mm × 2.1 mm); the column temperature was 40 °C. The mobile phase A was 2 mmol/L ammonium formate, and the mobile phase B was acetonitrile; gradient program: 0–2 min, 10% A; 2–4 min, 10%–50% A; 4–5 min, 50% A; 5–5.2 min, 50%–10% A; 5.2–7 min, 10% A. Sample volume: 10 μL; low rate: 0.4 mL/min.

Mass spectrometry conditions: The positive ion ionization (ESI^+^) from the electrospray ion source was used to select quantitative ions (127.1 > 85.2) and qualitative ions (127.1 > 68.4). The capillary voltage was set to 3500 V, and the ion source temperature was 350 °C. The pyrolysis voltage was 77 V; the collision energy m/z 127.1 > 85.2 was 17 V, and the collision energy m/z 127.1 > 68.4 was 28 V.

Sample pretreatment of β-agonists: Samples (2 g, accurate to 0.01 g) were weighed in a 50 mL centrifuge tube. After 8 mL of sodium acetate buffer was added and fully mixed, 50 μL of β-glucuronidase/arylsulfatase was added for mixing. Then, the samples were hydrolyzed in a water bath at 37 °C for 12 h. Subsequently, 100 μL of 10 ng/mL internal standard solution was added to these samples. The cover was oscillated in a horizontal oscillator for 15 min and then centrifuged at 5000 r/min for 10 min. After that, 4 mL of supernatant was transferred to another 50 mL centrifuge tube, followed by the addition of 5 mL of 0.1 mol/L perchloric acid solution and even mixing. The pH was adjusted to 1 ± 0.3 with perchloric acid. After the centrifugation at 5000 r/min for 10 min, all the supernatant (about 10 mL) was transferred to another 50 mL centrifuge tube, and the pH was adjusted to 11 with 10 mol/L sodium hydroxide solution. Next, the mixed solution containing 10 mL of saturated sodium chloride solution and 10 mL of isopropyl alcohol-ethyl acetate (6 + 4) was added for full extraction, followed by the centrifugation at 5000 r/min for 10 min. All organic phases were transferred and dried with nitrogen in a water bath at 40 °C. After that, 5 mL of sodium acetate buffer solution was added for ultrasonic mixing so that the residue could be fully dissolved for later use.

Chromatographic conditions: The chromatographic column was the Kinetex LC Column C_18_ (2.6 μm, 100 mm × 2.1 mm); the column temperature was 30 °C; the mobile phase A was 0.1% formic acid water, and the mobile phase B was acetonitrile; gradient program: 0–1 min, 97% A; 1–3.5 min, 97–70% A; 3.5–4.5 min, 70% A; 4.5–6.4 min, 70–25% A; 6.4–6.5 min, 25–5% A; 6.5–8.2 min, 5% A; 8.2–8.3 min, 5–97% A; 8.3–10 min, 97% A. Sample volume: 10 μL; flow rate: 0.4 mL/min.

Mass spectrometry conditions: The electrospray ion source (ESI^+^) was used at the ion source temperature of 500 °C, a capillary voltage of 5500 V and an atomizer flow rate of 55 L/min. The mass spectrometry parameters of three β-agonists are listed in Table 1.

## 3. Results and Discussion

This section may be divided by subheadings. It should provide a concise and precise description of the experimental results and their interpretation, as well as the experimental conclusions that can be drawn. These microspheres were characterized by Fourier transform infrared spectroscopy (FTIR), scanning electron microscopy (SEM), transmission electron microscopy (TEM), particle size analysis and saturation magnetization measurement.

### 3.1. Scanning Electron Microscopy

The structure of magnetic polystyrene microspheres was characterized by scanning electron microscopy (SEM) with an FEI QUANTA FEG250 from the United States. Figure 1 presents the transmission electron microscope diagram of these magnetic polystyrene microspheres. It can be seen from the figure that these microspheres are mostly perfectly spherical, with the most common particle size being 10 μm; these microspheres also have certain dispersion. Some smaller particles and irregular substances may be caused by unwrapped or self-aggregated polystyrene. The smooth surface of magnetic microspheres provides a good condition for further functionalization.

### 3.2. Transmission Electron Microscopy

The structure of magnetic polystyrene microspheres was characterized by transmission electron microscopy (TEM) with an FEI Tecnai F20. Figure 2 presents the transmission electron micrographs of magnetic polystyrene microspheres. It can be seen from the figure that Fe_3_O_4_ nanoparticles are formed by many nanocrystal clusters, with a spherical shape and a rough surface. The particle size is about 200 nm, and the dispersion uniformity is good. After polystyrene is coated on the surface of Fe_3_O_4_ particles, the particle size of Fe_3_O_4_@PS increases to 500 nm. Compared with Fe_3_O_4_ nanoparticles, a layer of spherical particles is added on the surface, which can effectively prevent the oxidation of Fe_3_O_4_ in air. In addition, introducing the polystyrene layer enhances the adsorption capacity and biocompatibility of the material.

### 3.3. Infrared Spectrum Analysis

The structures of magnetic nano-microspheres, magnetic polystyrene microspheres and functionalized magnetic polystyrene microspheres were characterized by a Thermo Fisher Nicolet -iS5 infrared spectrometer, with KBr as the substrate. It can be seen from Figure 3a that 560 cm^−1^ is the Fe=O stretching vibration of magnetic nano-microspheres. Compared with Figure 3a, Figure 3b presents three additional peaks in the skeletal vibration frequencies of benzene ring at 1445 cm^−1^, 1488 cm^−1^, and 1594 cm^−1^, respectively. The absorption peak at 3024 cm^−1^ is attributed to the C-H stretching vibration of the benzene ring. There are two strong absorption peaks at 696 cm^−1^ and 754 cm^−1^, respectively, which proves the presence of styrene. There is a stronger absorption peak at 1718 cm^−1^, which is the characteristic absorption peak of C=O. There is a broadened absorption peak at 3425 cm^−1^, which is the stretching vibration frequency of OH; this may be caused by the moisture contained in the sample. The absorption peak at 538 cm^−1^ is presumed to be the absorption peak of Fe_3_O_4_ powder, which is inconsistent with that at 560 cm^−1^ reported in a related article [36]. It may be caused by the particle size of Fe_3_O_4_ powder in the micron scale; hence the deviation is induced. The obtained products were characterized by infrared spectroscopy. Compared with Figure 3b, Figure 3c presents three additional new peaks, including a broad peak at 3400 cm^−1^, which is induced by the vibration of the water molecules adsorbed by the sulfonic acid group, and two peaks at 1164 cm^−1^ and 1001 cm^−1^, which may result from the C-S bond. This indicates the introduction of the sulfonic acid group on the benzene ring of the product. Moreover, the absorption peak of the C-H bond of the benzene ring at 1687 cm^−1^ is significantly enhanced. The higher the sulfonation degree, the wider the absorption band.

### 3.4. Saturation Magnetization Analysis

The saturation magnetization of these magnetic polystyrene microspheres was measured by an MPMS-3 VSM magnetometer. Figure 4 presents the hysteresis curves of Fe_3_O_4_ and Fe_3_O_4_@PS. As shown in the figure, the hysteresis loops of Fe_3_O_4_ and Fe_3_O_4_@PS all pass through the origin, indicating that the above materials have no hysteresis and are superparamagnetic at room temperature. The saturation magnetization (Ms = 65.524 emu/g) of Fe_3_O_4_ is significantly higher than that of Fe_3_O_4_@PS (Ms = 9.962 emu/g). This indicates that the saturation magnetization decreases continuously with the gradual increase in non-magnetic materials outside the magnetic core, which results in the weakening of magnetism. During further functionalization, the saturation magnetization of these magnetic polystyrene microspheres decreases to Ms = 4.267 emu/g with the increase of surface sulfonic groups. However, Fe_3_O_4_@PS and Fe_3_O_4_@SPS can still be easily separated from the solution under an external magnetic field, which meets the requirements of pretreatment experiments.

### 3.5. XRD Result Analysis

Fe_3_O_4_, magnetic polystyrene microspheres and sulfonated magnetic polystyrene microspheres were analyzed by an X-ray diffractometer (Bruker D2 Phase, Germany). It can be seen from Figure 5 that diffraction peaks with different peak intensities still appear at 2θ = 30.121°, 35.498°, 43.205°, 53.496°, 57.022°, and 62.603° for Fe_3_O_4_ nanoparticles and prepared magnetic polystyrene microspheres, which is consistent with the position of characteristic diffraction peaks of Fe_3_O_4_. Thus, Fe_3_O_4_ nanoparticles could not destroy the crystal structure of ferrite during modification. It also shows that there is no change in the crystal structure of ferrite during suspension polymerization. In addition, the diffraction peak of Fe_3_O_4_@PS magnetic microspheres is weaker than that of Fe_3_O_4_ nanoparticles, which is mainly caused by the weakening of the diffraction peak intensity of Fe_3_O_4_ due to the presence of polystyrene. The large diffraction dispersion peak at 2θ = 20° is a typical amorphous polystyrene diffraction peak. The diffraction peak of Fe_3_O_4_@SPS magnetic microspheres is weaker than that of Fe_3_O_4_@PS, indicating that the magnetism of ferrites continues to be weakened by further functionalization of chlorosulfonic acid. However, the separation effect would not be affected.

### 3.6. Organic Element Analysis

The reaction time of Fe_3_O_4_@PS microspheres with 1:1 molar ratio of chlorosulfonic acid and concentrated sulfuric acid was investigated at 25 °C with mechanical stirring, and the relationship between the reaction time and the loading capacity of the two methods was studied. The reaction speed of the two sulfonation methods was very fast, and the loading amount increased rapidly with the increase in time; the magnetic microspheres reached the maximum loading amount in 5 min. After 5 min, the loading amount did not change much with the increase in reaction time. On the whole, the sulfonation effect of the chlorosulfonic acid method (maximum loading of 3.115 mmol/g) was better than that of the concentrated sulfuric acid method (maximum loading of 2.185 mmol/g).

The elemental composition of Fe_3_O_4_@SPS microspheres was determined by elemental analysis. After modification, there was no sulfur element in the magnetic polystyrene microspheres. Then, 10 mg of Fe_3_O_4_@SPS microspheres were weighed and the contents of C, N, H and S were determined. The loading amount of microspheres was calculated according to the following formula. The average loading amount of microspheres was found to be 3.115 mmol/g by parallel testing of three groups of samples. X represents the mass percentage of sulfur; M_0_ represents the molar mass of sulfur (g/mol).
L_A_ = X/M_0_ × 100%

### 3.7. Adsorption Performance Results of Actual Samples

Adsorption effect analysis of melamine samples: The melamine standard was prepared with acetonitrile to 1000 μg/L and stored at −18 °C. The standard solution was diluted to 10 μg/L, 20 μg/L, 50 μg/L, 100 μg/L, 200 μg/L and 300 μg/L standard working solutions with formic acid aqueous solution, followed by oscillation and vortex. The standard curve was plotted according to the test results of HPLC-MS/MS, with the results shown in Figure 6a.

Adsorption effect analysis of β-agonists: Mixed standard solution: An appropriate amount of single standard stock solutions of clenbuterol hydrochloride, ractopamine and salbutamol were taken and diluted with methanol to 1000 μg/L mixed standard solutions. These solutions were stored in the dark at −18 °C. Mixed working solution of isotope internal standard: An appropriate amount of single-standard internal standard storage solutions of clenbuterol-D9, ractopamine-D6 and salbutamol-D3 isotopes were, respectively, taken and diluted with methanol to 1000μg/L mixed isotope internal standard solutions. These solutions were stored in the dark at −18 °C. The mixed standard solutions were diluted with acetonitrile to obtain standard working solutions with external standard concentrations of 1 μg/L, 2 μg/L, 5 μg/L, 10 μg/L, and 20 μg/L (all isotope internal standards were 10 μg/L). The standard curve was plotted according to the test results of HPLC-MS/MS, with the results shown in Figure 6b.

The blank sample milk was used as the matrix for the standard addition recovery experiment. The standard addition concentrations were 20 μg/L, 50 μg/L, 100 μg/L, 150 μg/L, and 200 μg/L, respectively. The spiked samples and blank samples were tested after pretreatment, with the results listed in Table 2. It can be seen from the table that the adsorption amount of melamine on Fe_3_O_4_@SPS microspheres reaches about 55% at the spiked concentration of 20 μg/L. In addition, the recovery rate increases with the increase in the concentration of melamine. This indicates that there is a remarkable adsorption effect of Fe_3_O_4_@SPS microspheres on melamine.

After 8 blank matrix samples of pork, mutton, and meat pine in 2 g were weighed and placed in a 50 mL centrifuge tube, clenbuterol hydrochloride, salbutamol and ractopamine standard solutions were added to 6 samples to reach a concentration of 0.5 μg/kg. The remaining 2 samples were used as blank matrix samples which were tested on the machine. Each centrifuge tube was added with 10 mg of Fe_3_O_4_@SPS, in turn, followed by oscillation and vortex at 25 °C for 24 h. The supernatant was separated by strong magnetic separation and determined by HPLC-MS/MS. The test results are listed in Table 3.

It can be seen from the table that the recovery rate of clenbuterol hydrochloride, salbutamol and ractopamine in pork, mutton, and fried pork flakes were 92–102%, 90–100% and 94–104%; 94–102%, 90–102% and 90–98%; 90–102%, 90–98% and 92–102%, respectively. All results meet the requirements of GB/T 27404-2008 *Criterion on Quality Control of Laboratories-Chemical Testing of Food* (60–120%). The relative standard deviations of clenbuterol hydrochloride, salbutamol and ractopamine in pork, mutton, and pine were 3.6%, 4.1% and 3.9%, respectively. 2.8%, 5.8% and 3.3%; 4%, 3.0% and 4. 3%, respectively. The relative standard deviation (coefficient of variation) also meets the requirements of GB/T 27404-2008 *Criterion on Quality Control of Laboratories-Chemical Testing of Food* (less than 30%). This indicates that Fe_3_O_4_@SPS microspheres have a very significant adsorption effect on β-agonists.

## 4. Conclusions

In this study, the magnetic polystyrene microspheres with good morphology were prepared by improved microemulsion polymerization with nano-sized magnetic microspheres Fe_3_O_4_ as carriers, and styrene, divinylbenzene and triallyl isocyanate as polymerization monomers. The Fourier transform infrared spectroscopy results showed that there were Fe_3_O_4_ nanoparticles, styrene and a sulfonic acid group in these microspheres. The scanning electron microscopy (SEM) and transmission electron microscopy (TEM) results showed that these magnetic polystyrene microspheres had good sphericity and the average particle size was less than 10μm. The vibrating sample magnetometer (VSM) results showed that the saturated magnetization of magnetic polystyrene microspheres was 3.225 emu/g, which was superparamagnetic. The X-ray diffraction (XRD) spectrum analysis results showed that Fe_3_O_4_ nanoparticles would not destroy the crystal structure of ferrite during modification, and would also not change the crystal structure of ferrite during suspension polymerization. At the same time, typical amorphous polystyrene diffraction peaks appeared at 2θ = 20°.

Moreover, these magnetic polystyrene microspheres were functionalized by chlorosulfonic acid. The infrared spectrum analysis results showed that the magnetic polystyrene microspheres were successfully grafted with sulfonic acid groups. The results, based on an organic element analyzer, showed that the loading amount of Fe_3_O_4_@SPS microspheres was 2.185 mmol/g. The actual sample test and HPLC-MS/MS results showed that Fe_3_O_4_@SPS microspheres had strong adsorption properties for melamine, clenbuterol, ractopamine and salbutamol.

In summary, based on the work of Jin et al. we obtained satisfactory magnetic composite microspheres by optimizing the ratio of initiator and magnetic microspheres, the reaction time and the pH of the reaction system as a prerequisite for the experiments conducted in this paper. By reviewing the literature and comparing the experiments, the selection of a sulfonation reagent, the control of sulfonation time and the adjustment of the ratio, we finally decided to choose chlorosulfonic acid as the sulfonation agent; the molar ratio of chlorosulfonic and magnetic composite microspheres was 1:1, and the optimal sulfonation time was 5 min.

## Data Availability

Data is contained within the article.

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
