# Peer review of "Preparation of Functionalized Magnetic Polystyrene Microspheres and Their Application in Food Safety Detection"

_polymers, 2022, doi:10.3390/polym15010077_

Round 1

Reviewer 1 Report

1. The author claims that the obtained composite can be used for food detection, however, the author performed only an adsorption test. How about the specific detection and selectivity test?

2. Please add a scale bar for the TEM image. 

3. For FTIR, why do the FTR peaks around 3000 disappear after sulfonation? 

4. Please give all peak assignments for FTIR peaks. 

5. How does the magnetism affect the crystal structure, which was mentioned in the XRD section? 

6. Why do the main characteristic XRD peaks of Fe3O4 disappear in Fe3O4@spc? Is the sulfonation affect the crystal of Fe3O4?

Reviewer 2 Report

The authors report about the preparation of functionalized magnetic polystyrene microspheres. Comments are provided below.

Introduction

- The authors list possible applications of microspheres, including targeted drug delivery. If the authors relate this item specifically to magnetic hyperthermia, it is not accurate since the latter utilizes core-shell structures in the nano scale. Please clarify this point.

- It is not clear why polystyrene is the relevant material for this study. In other words: which other organic materials would be comparable to it and why polystyrene is better than them?

Materials and Methods

- What is the ratio implemented in the mixture between sodium dodecyl sulfate and PEG4000?

- Scheme 1 is apparently not called in the section. Please check it.

- The authors state that the use of chloride sulfonic acid as a sulfonating agent has some advantages; please specify compared to what/under which conditions.

- What is the concentration of oleic acid-ethanol solution? Please specify it because otherwise others cannot really reproduce reported results.

- Please specify the certain amount of chloride sulfonic acid added in mechanical stirring.

- Scheme 2 is apparently not called. Please check it.

Results and Discussion

- Figure 2 is confusing. In both images the spheres seem the same dimensions, and it is not clear to differentiate the scale bar. Please clearly indicate in the images the length of the scale bar.

- Please rewrite the sentence "... frequencies of the benzene ring, which indicates the presence of benzene ring". Redundant benzene ring.

- In the sentence "... with that at 560cm-1 277 reported in a related article." please add the reference of the related article.

- Figure 4: the image, in the scale presented, does not guarantee that the particle are superparamagnetic. In fact, it is needed to guarantee that during measurement, zero field applied really means zero field. Please provide an inset of the origin in Figure 4 and indicate how the measurement system was calibrated.

- The decrease in the saturation should be proportional to the volume fraction of magnetic phase reduction in the core shell structure. Can the authors provide an estimation of how much coating was applied to the particles and compare if the results are within the expected range?

- The sentence "Fe3O4 nanoparticles could not destroy the crystal structure of ferrite during modification" is unclear. Please clarify it.

- If the XRD broad peak at 20 is expected, why the one from the blue diffractogram in Figure 5 is different than that of the red?

- In Figure 6 please add error bars to the experimental data.

Reviewer 3 Report

This manuscript presents the preparation of polystyrene-covered magnetic nanoparticles functionalized with sulfonic acids.

The authors claim that the preparation method for the polystyrene-covered magnetic nanoparticles has been improved and that their functionalization has also been improved.

The functionalized nanoparticles were used to determine veterinarian drug residues in food.

The paper is interesting, but the writing is somewhat confusing.

Some aspects need to be modified for this manuscript to be publishable.

The authors don’t indicate how the Fe3O4 nanoparticles were prepared. At least the general methodology should be indicated (coprecipitation, thermal decomposition…)

In the description of the methods, the improvements are not mentioned. What is new in the polymerization protocol and the functionalization?

The description of the procedures is incomplete. In the functionalization of the microspheres, it says, “a certain amount” and “a certain temperature”. With those indications, the paper cannot be reproduced.

Chloride sulfonic acid should be chlorosulfonic acid.

Ammonium formic acid should be ammonium formate.

In Table 1, the terms “parent ion” and “daughter ion” have been deprecated (IUPAC recommendations 2013) Precursor ion and product ion should be used instead.

Also in Table 1, what do the stars in the third column means?

Table 2 is not clear. Is the “result of the survey” not related to the concentration of the sample?

Also, some numbers are missing (the mean value for the 200 ug/L concentration needs another result)

In the text, it is said that the recovery rate decreases with the increase of the concentration, but from Table 2 it seems to be the opposite.

Some references are missing, for instance, in line 141 for the three sulfonation methods mentioned some references should be included.

Some of the references included are difficult to find. Thus, references 13, 20, 25, 26, 28, 30, and 31 do not show up in a standard search.

In the references section, the formatting is not coherent. DOI is missing in certain articles.

The conclusions are too general, with just a list of the research done. The improvements and goals achieved should be emphasized. 

Round 2

Reviewer 1 Report

The authors have addressed all of the concerning points. Thus, It can be accepted for publication. 

Author Response

Thank you for your kind instructons and decision.

Reviewer 2 Report

The authors addressed initially all comments.

Concerning Figure 4, it is still not clear whether the particles are superparamagnetic (even at room temperature). Only by this image is not possible to conclude that for different reasons (image scale, no inset demonstrating really zero coercivity). Also the XRD data could be used to estimate the dimension of the magnetic particles: which dimension would be found?

Also  the scale in Figure 2 is virtually impossible to be observed even in the PDF file. Please address that.

Reviewer 3 Report

After the modifications introduced by the authors, this manuscript is now ready to be published

Author Response

(The authors gave the same response as above.)

Round 3

Reviewer 2 Report

N/A